# Functional Profiling of Soft Tissue Sarcoma Using Mechanistic Models

**DOI:** 10.3390/ijms241914732

**Published:** 2023-09-29

**Authors:** Miriam Payá-Milans, María Peña-Chilet, Carlos Loucera, Marina Esteban-Medina, Joaquín Dopazo

**Affiliations:** 1Computational Medicine Platform, Andalusian Public Foundation Progress and Health-FPS, 41013 Sevilla, Spain; miriam.paya@juntadeandalucia.es (M.P.-M.); maria_pena@iislafe.es (M.P.-C.); carlos.loucera@juntadeandalucia.es (C.L.); marina.esteban@juntadeandalucia.es (M.E.-M.); 2Centro de Investigación Biomédica en Red de Enfermedades Raras (CIBERER), FPS, Hospital Virgen del Rocío, 41013 Seville, Spain; 3Institute of Biomedicine of Seville, IBiS/University Hospital Virgen del Rocío/CSIC/University of Sevilla, 41013 Sevilla, Spain; 4FPS/ELIXIR-ES, Fundación Progreso y Salud (FPS), CDCA, Hospital Virgen del Rocío, 41013 Sevilla, Spain

**Keywords:** soft tissue sarcoma, mechanistic models, signaling pathways, therapeutic targets RNA-seq, transcriptome, profiling

## Abstract

Soft tissue sarcoma is an umbrella term for a group of rare cancers that are difficult to treat. In addition to surgery, neoadjuvant chemotherapy has shown the potential to downstage tumors and prevent micrometastases. However, finding effective therapeutic targets remains a research challenge. Here, a previously developed computational approach called mechanistic models of signaling pathways has been employed to unravel the impact of observed changes at the gene expression level on the ultimate functional behavior of cells. In the context of such a mechanistic model, RNA-Seq counts sourced from the Recount3 resource, from The Cancer Genome Atlas (TCGA) Sarcoma project, and non-diseased sarcomagenic tissues from the Genotype-Tissue Expression (GTEx) project were utilized to investigate signal transduction activity through signaling pathways. This approach provides a precise view of the relationship between sarcoma patient survival and the signaling landscape in tumors and their environment. Despite the distinct regulatory alterations observed in each sarcoma subtype, this study identified 13 signaling circuits, or elementary sub-pathways triggering specific cell functions, present across all subtypes, belonging to eight signaling pathways, which served as predictors for patient survival. Additionally, nine signaling circuits from five signaling pathways that highlighted the modifications tumor samples underwent in comparison to normal tissues were found. These results describe the protective role of the immune system, suggesting an anti-tumorigenic effect in the tumor microenvironment, in the process of tumor cell detachment and migration, or the dysregulation of ion homeostasis. Also, the analysis of signaling circuit intermediary proteins suggests multiple strategies for therapy.

## 1. Introduction

Sarcomas are a group of rare heterogeneous malignancies that arise from cells of the mesenchymal lineage, including connective tissue, muscle, fat, bone, and cartilage [1]. These very rare neoplasms have an overall annual incidence of 5.6 cases per 100,000 adults in Europe, which represents only 1–2% of all cancers in adults [2]. According to the latest WHO classification, the heterogeneity of sarcomas embraces over 70 histological subtypes [3], making them diagnostically challenging. Since appropriate treatment and prognostication rely on an accurate diagnosis, the classification of soft-tissue tumors is increasingly shifting from the analysis of histological characteristics toward the identification of specific molecular features and mechanisms underlying carcinogenesis and disease progression [3,4]. In a simplistic approach, sarcomas are classically divided into two main groups: (1) Sarcomas with a single underlying genomic abnormality and (2) sarcomas with complex genomic alterations [5,6]. Although this classification is oversimplifying, over 50% of soft-tissue sarcomas fall into the complex group, highlighting the importance of deciphering the mechanisms of pathogenesis to improve clinical management [6]. 

At a molecular level, sarcomagenesis is driven by genetic and epigenetic alterations, including mutations, copy number changes, and gene expression alterations, which result in the activation of oncogenes or the loss-of-function of tumor suppressors, leading to uncontrolled cell proliferation. Mutations in sarcomas can occur in a variety of genes, including TP53, CDKN2A, CTNNB1, RB1, PTEN, and members of the RAS/RAF/MAPK pathway [4]. Somatic mutations in TP53, a tumor suppressor gene, account for up to 50% of sarcomas and are associated with a poor prognosis [7]. The loss-of-function somatic as well as germline alteration of CDKN2A, which encodes proteins involved in cell cycle regulation, is also associated with sarcomagenesis [8].

To understand the complex landscape of functional strategies that cells use to initiate their malignization and further progress to different sarcoma subtypes, a strategy based on mathematical modeling of the molecular cancer mechanisms has been used. Specifically, mathematical mechanistic modeling of cell signaling pathways provides a causal link between variation occurring at the level of gene activity (transcriptional activity) or integrity (mutational profile) and the observed phenotype diversity (at the level of cells, tissues, or organisms) [9]. Actually, such models have successfully been applied to reveal specific molecular mechanisms accounting for different diseases, such as cancer, diabetes, or Fanconi anemia [9,10,11], and to suggest personalized therapeutic interventions [12]. Moreover, mechanistic models offer a precise framework for simulating the signal transduction process, establishing a direct link between causal factors (gene expression levels) and resulting outcomes (signaling activity) [9,13], thereby conveying a sense of causality. Therefore, mechanistic models can also be used to predict the consequences of interventions [14,15], such as the effects of targeted drugs [16,17], combinatorial drug consequences [18,19], and even therapy responses [20,21]. Some successful applications of these models were the prediction of gefitinib and afatinib as new potential treatments for Fanconi anemia [10], later validated [22], or the prediction of repurposable drugs like SARS-CoV-2 [17], whose activity was proven by real-world data analysis for some of them [23,24].

This study elucidates the impact of general and specific processes in soft tissue sarcoma samples gathered from the Cancer Genome Atlas (TCGA) project [25], which contribute to patient prognosis and exhibit deviations from non-diseased sarcomagenic tissues. Leveraging the mechanistic modeling implemented in the HiPathia application [13,26], signal transduction circuit activities were assessed. Transcription Factor Target Enrichment Analysis [27] provides extra support for the findings. The results provide comprehensive insights into sarcoma biology and potential therapeutic avenues.

## 2. Results

### 2.1. Samples

Selected data from GTEx tissues and the SARC project were downloaded. Samples from GTEx on putative sarcomagenic tissues of mesenchymal origin were: 1293 adipose, 520 fibroblasts, 881 skeletal muscle, 553 esophagus muscularis, 384 stomach, 159 uterus, and 946 vessels. From TCGA, the 206 sarcoma samples include 7 subtypes: 50 dedifferentiated liposarcomas (DDLPS), 80 leiomyosarcomas (divided into 53 soft tissue STLMS and 27 uterine ULMS), 5 malignant peripheral nerve sheath tumors (MPNST), 17 myxofibrosarcomas (MFS), 10 synovial sarcomas (SS), and 44 undifferentiated pleomorphic sarcoma (UPS). 

Downloaded gene read counts from these samples were preprocessed together with TMM normalization and used for the estimation of activity values with HiPathia. Data exploration by dimensionality reduction with t-SNE and clustering showed a strong correlation of samples by the project (Appendix A). On sarcoma, most LMS samples, divided by uterine or soft tissue LMS and SS samples, formed close clusters.

### 2.2. Signal Transduction Circuit Activity Estimation

A common assumption in genome-scale modeling methodologies is considering RNA-Seq counts as trustable proxies for protein abundance and, consequently, protein activity [9]. This enables the modeling methods to infer the signal transduction intensities along signaling pathways from the gene expression measurements. For this purpose, HiPathia, an application that implements mechanistic models to recode gene expression values into the activity of signaling pathways, defined in the Kyoto Encyclopedia of Genes and Genomes (KEGG) in the pathways repository [28], was used. For this analysis, a total of 78 physiological signaling pathways were selected. Within these pathways, only 13 genes (0.5% of the total number of genes) were found to have missing expression values. For these few specific cases, the values are imputed based on the average expression of the dataset. HiPathia defines individual signaling circuits within the pathways to individualize functional activities. Each circuit is composed of nodes connected by activating or inhibiting relationships and ends in a single effector node, which is responsible for the cellular functions triggered by the circuit. A number of these functions can be ascribed to cancer hallmarks [29] with the CHAT tool [30,31], as described in the Methods section. A total of 1098 signaling circuits were identified by HiPathia within the 78 physiological KEGG pathways. 

Exploration of sample clustering with either t-SNE or a heatmap did not reveal significant differences compared to sample distances using expression values (see Figure 1), indicating that this method preserves the biological information of samples.

### 2.3. Survival

The activation of signaling pathways is closely related to cell fate and may have a causal influence on patient survival. Here, the association of the modeled circuit activity values with sarcoma patients’ overall survival was explored in detail. To achieve this, Cox proportional hazard models and Kaplan–Meier plots (Appendix A) were employed. Thirteen signaling circuits significantly associated with sarcoma patient survival were found (Table 1 and Appendix A). These circuits are part of pathways that are well-known drivers of tumorigenesis, such as NF-kB, RB1, or Akt, and other pathways related to the immune system. Pathways involved in the immune system (FcεRI signaling and platelet activation) are predicted to have a protective role in patient outcomes. The cell cycle pathway with the tumor suppressor retinoblastoma as a signaling circuit effector (*RB1*, see Table 1) has a coherent low hazard ratio. Conversely, the axon guidance pathways associated with cell motility, where the focal adhesion kinase (*PTK2*) acts as an effector, exhibit a high hazard ratio. However, signaling pathways may have multiple roles in tumorigenesis, which will be further discussed.

### 2.4. Transcription Factor Activation

Given the low expression levels exhibited by transcription factors (TF), the observed gene expression values in transcriptomic experiments may not represent their appropriate activity in the cell. A workaround to indirectly estimate the activity of a TF is to assess the activity of its target genes using any enrichment method. Here, the TFTEA [27] method was used. A total of 130 TF targets along the circuits associated with survival were found, 65 of which were dysregulated. The results indicate that the most shared transcription factors by sarcoma subtypes have target genes overrepresented in the overexpressed part of the rank (Table 2 and Appendix A). *SPI1* (a.k.a. PU.1) is the TF whose targets are present in most of the circuits. In particular, it has 41 targets within the signaling circuits whose expression is dysregulated, including *PIK3CG* and *AKT2*, central components of the PI3K/Akt signaling network present in various pathways, as well as components of the MAPK signaling network such as *IKBKB, MAPK10,* or *RAC2. IKZF1* (Ikaros protein) targets participate in the protective circuits of the Fc epsilon signaling pathway, platelet activation, and HIF-1 signaling pathway. Among Ikaros targets, *Lyn kinase* and *PRKCB* (protein kinase C, beta subunit) were found to be upregulated. *Lyn* participates in the activation of PI3K/Akt networks within the indicated pathways, while it activates *PKC* by downstream calcium signaling.

### 2.5. Differential Signaling

To assess the altered cellular functions of sarcoma, the signaling activities of each sarcoma subtype were contrasted with those of their putative sarcomagenic normal tissues. In addition to some common deregulated signaling circuits, each sarcoma subtype also exhibited its own unique set of deregulated circuits. Functional annotation of the circuits that are dysregulated by each sarcoma subtype primarily highlights DNA replication, apoptotic processes, responses to cytokines, and Fc-receptor signaling (Figure 2 and Appendix A). Overall, upregulation of immune-related circuits or anti-apoptotic circuits, such as those with effector Bcl-2, and downregulation of apoptotic or growth-inhibitory processes (Appendix A) were found. Interestingly, there is a set of circuits that simultaneously displayed significant differential activation in all subtypes (Table 3 and Appendix A). These belong to the following pathways: adipocytokine signaling pathway, aldosterone synthesis and secretion, Fc gamma R-mediated phagocytosis, PI3K-Akt signaling pathway, and focal adhesion. Multiple components of dysregulated circuits are also affecting survival-related circuits since central components of signaling are shared and intertwined in a complex network. The potential implications of these circuits on the onset and progression of sarcomagenesis will be further discussed below.

### 2.6. Hallmarks of Cancer

Differentially activated signaling circuits were associated with hallmarks of cancer using CHAT. With this classification, the abundance of such hallmarks on the dysregulated circuits for each sarcoma subtype was assessed. The most frequently dysregulated processes are related to cellular energetics, invasion/metastasis, and the avoidance of the immune system. At the same time, less frequently, they are involved in enabling replicative immortality, evasion of growth suppressors, and genomic instability and mutation (Figure 3). In consonance with the observed hallmarks, various sarcoma subtypes commonly exhibit dysregulation in processes related to the immune system, including Fc gamma R-mediated phagocytosis. Additionally, they demonstrate dysregulation in metabolic pathways like the adipocytokine signaling pathway and processes related to cell motility, such as the focal adhesion pathway.

## 3. Materials and Methods

### 3.1. Data Download and Pre-processing

Raw gene counts for tumor and normal tissue samples were obtained from the recount3 R package [33]. Recount3 provides RNA-Seq dataset counts where raw reads from multiple sources, including cancer and non-diseased tissues generated by the TCGA and the Genotype-Tissue Expression Project (GTEx), respectively, were uniformly processed with a pipeline using STAR [34] and Megadepth [35] to align and quantify the reads to the UCSC hg38 assembly. Tumor samples were obtained from the SARC project, belonging to the TCGA consortium, while normal samples were downloaded from multiple tissue projects of the GTEx consortium [36] and further selected for specific tissue site details (Table 4).

Low-quality and non-primary tumor samples from the sarcoma project were filtered out. The subtype classification was modified according to previous expert pathology reviews [37]. We kept genomic features mapped to an Entrez ID on the org.Hs.eg.db Bioconductor package v3.12.0 [38].

The downloaded raw counts for all samples were normalized using the Trimmed Mean of M-values (TMM) method [39], as implemented in the edgeR package [40] version 3.32.1, which computes scaling factors assuming that most genes are not differentially expressed. Normalized counts in the log scale were obtained with the *voom* function provided in the limma package version 3.46.0 [41], which also models the mean–variance relationship of normalized values with precision weights suitable for the analysis of differential gene expression. 

### 3.2. Mechanistic Modeling of Human Signal Transduction

The mechanistic model replicates the dynamics of the signaling circuits defined within pathways. These circuits are depicted as directed graphs, establishing connections from receptor to effector proteins through a series of activations and inhibitions mediated by intermediate proteins. Ultimately, effector proteins at the end of these circuits initiate distinct cellular functions. Malfunctions of these circuits can trigger cancer hallmarks in neoplastic cells. The mechanistic model emulates signal transduction along these circuits by considering protein activity levels. In this context, gene expression values serve as proxies for the presence of their corresponding active proteins [42,43,44,45]. Consequently, for a circuit to become active and effectively transmit the signal triggering a specific function, it necessitates the concurrent presence of the entire chain of proteins connecting the receptor to the effector while ensuring the absence of inhibitory proteins that might impede signal propagation along the circuit. Irrespective of the circuit’s topology, the signal propagates through the nodes within it based on the subsequent recursive rule for each node:(1)$$S_{n}=\upsilon_{n}\cdot \left(1-\prod_{s_{a}\in A}\left(1-s_{a}\right)\right)\cdot \prod_{s_{i}\in I}\left(1-s_{i}\right)$$
where the signal intensity (*Sn*) for the current node (*n*) is determined by its normalized gene expression value (*v_n_*). This determination is influenced by the sets of activation signals (*s_a_*) from the set of activation edges (*A*) and inhibitory signals (*s_i_*) from the set of inhibition edges (*I*), as detailed in [13]. This modeling strategy ensures that causality is considered in a comprehensive context, where gene expression levels dictate the ultimate functional outcomes. These outcomes are initiated by effector proteins, following the activation and inhibition rules governed by the relationships among the proteins within the signal circuit. Consequently, alterations in node activity will manifest or go unnoticed, depending on the specific circuit’s topology.

### 3.3. Calculation of Pathway Activity

Prior to pathway activity computation, normalized values were rescaled to the range 0–1 using the function *normalize_data* from the package HiPathia 2.6.0 [46]. HiPathia (HIgh throughput PATHway Interpretation and Analysis) is designed to transform gene count data into the activity of signaling pathways divided into single-effector circuits that drive specific cell functions [13,26]. Human physiological pathways were provided to the *hipathia* function. The function *normalize_paths* was used to account for the length of the circuits in the activity calculation.

Visualization of normalized counts and activity values was performed using the t-distributed stochastic neighbor embedding method (t-SNE) for visualization of the structure of high-dimensional data [47] using the function provided in the package Rtsne [48] version 0.15, as well as on heatmaps using the pheatmap package [49] version 1.0.12 with the complete clustering method and, as distance measures, euclidean for features and correlation for samples.

### 3.4. Survival Analysis

Survival analysis was performed on sarcoma samples using the survival R package [50] version 3.2–10. From this package, Cox proportional hazards regression models [51] were computed to estimate differences in patient survival in association with circuit activity or gene expression values. To create the censored survival object, maximum values of “days to death” or “days to last follow-up,” with vital status as a censoring variable, were gathered from the clinical metadata of TCGA samples in Recount3. A model for each circuit was computed with the *coxph* function. The *p*-value of each model was adjusted for multiple tests with the Benjamini and Hochberg method (FDR, false discovery rate) [52]. In addition to Cox regression, the proportional assumption was tested with the *cox.zph* function of the survival package. Kaplan–Meier survival curves were constructed with the Survminer package version 0.4.9 [53] from selected patients with high and low circuit activity. High and low circuit activity level was defined based on scaling activity levels into z-scores and setting a cutoff of +/−0.5, corresponding to the 25% upper and lower percentile of the observed values, respectively. 

Additional survival analysis of circuit-integrating genes was achieved on the web application Kaplan–Meier plotter [54], where gastric cancer was selected as the closest option sarcoma. Upper to lower terciles were compared to obtain terciles values on hazard ratios, confidence intervals, and probability values. Additionally, the UALCAN [55] web application was also employed for survival analysis, specifically selecting the sarcoma project. Survival plots were generated comparing high/low expression samples, from which high/low hazard ratios are visualized with a probability value.

### 3.5. Differential Expression/Activity Analysis

For differential comparisons between sarcoma subtypes and normal tissues of both normalized expression and activity values, limma [41] was used. Limma pipeline consists of linear model fitting followed by standard error moderation with an empirical Bayes method. The model matrix included the tissue categories (sarcoma subtypes and normal tissues) and the tissue source site (centers where samples were collected) for correction as a batch effect. From these results, the set of differentially expressed genes (DEGs) and differentially activated circuits were obtained.

### 3.6. Transcription Factor Enrichment Analysis

The results of differential expression analysis were used for indirect estimation of transcription factor activity by enrichment analysis of their corresponding target genes using the Transcription Factor Target Enrichment Analysis (TFTEA) [27] tool. TFTEA carries out an enrichment analysis of targets for each transcription factor over a list of ranked DEGs using univariate gene set analysis with a logistic regression model. The set of transcription factor–target interactions used in this analysis was obtained from the papers describing TFTEA [27] and Dorothea, a database that collects transcription factors’ targets (regulons) [56]. From the Dorothea database, transcription factor–target interactions with support from at least two sources (confidence levels A, B, and C, as defined in [56]) were selected. Functional annotations of genes were gathered from the COSMIC Cancer Gene Census v96 [57] and the databases ONGENE [58] and TSGENE [59].

### 3.7. Functional Analysis

Gene ontology (GO) enrichment analysis was performed with the package enrichR version 3.0 [60]. Gene effectors of differentially activated circuits for each sarcoma subtype were used to estimate the enrichment of terms on the biological process ontology version 2021. Enrichment of differentially activated circuits in hallmarks of cancer [29] was achieved using a previous annotation of signaling circuit effectors to hallmarks performed with the Cancer Hallmarks Analytics Tool (CHAT) [30,31]. A CHAT value cutoff of 0.15, corresponding to a 95% percentile according to previous studies [31], was used to select hallmarks associated with each circuit.

## 4. Discussion

The activity of signaling pathways on sarcoma samples has been analyzed to, on the one hand, find how the degree of signaling pathway activity affects overall patient survival and, on the other hand, discover the alterations in sarcoma samples in contrast to sarcomagenic non-diseased tissues. In the present work, 13 circuits from 8 pathways associated with sarcoma patient survival, nine differentially activated circuits belonging to five pathways, and a set of eight enriched transcription factors have been reported.

Implementations of mechanistic models, such as HiPathia, provide a sophisticated and detailed interface for closely studying the functional state of cells using RNA-Seq data with high sensitivity [9,13]. In this study, mechanistic modeling of signaling activity was applied to publicly available data from TCGA and GTEx. This approach enabled the extraction of circuit activities based on KEGG signaling pathways. Analyzing signaling pathways through the HiPathia application provided a more accurate description of functions in sarcoma samples, such as growth, replication, transcriptional activity, and motility, which are relevant to tumor cell biology. The results align with current knowledge of cancer, highlighting the predictive power of mechanistic models in HiPathia to simulate real processes under specific cancer conditions. This capacity offers a valuable framework for evaluating therapeutic options. 

### 4.1. Roles in the Tumor Microenvironment

Overall, a protective role of an enhanced immune system was observed, supported by low hazard ratios in immune-related pathways such as the Fc epsilon RI signaling pathway with four significant circuits (Table 1). One of them involves the aggregation of the high-affinity IgE receptor Fc epsilon RI upon binding to the IgE antibody. Interestingly, expression of the three different subunits of the Fc epsilon RI has been associated with a good prognosis in various cancers, such as breast, lung, adenocarcinoma, osteosarcoma, and sarcoma (see Appendix A) [61,62,63]. Moreover, the significantly activated transcription factor *SPI1* could be used as a predictor together with *FCER1G* of the immune infiltration, suggesting a protective role for this TF [63]. The other three circuits significantly related to survival were different effector branches (with effector node proteins AKT—*AKT1–3*, JNK—*MAPK8–10*, and p38—*MAPK11–14*) from a common upstream receptor. Remarkably, these proteins activate transcription factors within mast cells and are ultimately involved in the production and release of cytokines and arachidonic acid [64,65,66], which is known to affect the tumor microenvironment (TME) and to promote vascularization and platelet aggregation, among other effects. Using these circuits, most sarcoma subtypes upregulate the intermediate proteins Lyn kinase, GAB2, or Vav proteins, which have dual roles in cancer progression [67,68,69]. Actually, Lyn is a target of the Ikaros protein IKZF1, which was shown to increase the immune infiltrate in solid tumors and enhance anti-tumor immunotherapy [70]. Another circuit uses a different signaling cascade to activate the cytosolic phospholipase A2 (cPLA2), involved in the production of arachidonic acid, to mediate the release of eicosanoids, active molecules that modulate inflammation. PLA2 is further activated on the “platelet activation pathway” through a PI3K/Akt cascade involving p38/ERK MAP kinases to produce arachidonic acid, which is sequentially modified by cyclooxygenase 1 (COX-1) and thromboxane A synthase 1 (TXS) to produce 11-dehydro-thromboxane B2, a vasoconstrictor involved in platelet aggregation. In sarcoma, platelet activation inhibits metastases and promotes tumor-suppressor genes [71]. The inducible isoform *COX-2* responds to TNF and p38/NF-κB activation and has been found to be protective against breast cancer [72]. In sarcoma, not only decreased hazard ratios for *COX* isoforms were observed, but also for the activation of the *COX-2* activator NF-κB through the “MAPK signaling pathway”.

An increased activation of the immune-related “Fc gamma R-mediated phagocytosis” pathway towards the activation of the Arp2/3 complex (Actin-related protein 2/3 complex) was also observed, as well as the previous node with *WAS/WASL* genes (encoding WASP and N-WASP nucleation-promoting factors). The heptameric Arp2/3 complex nucleates actin fibers into branched filaments that, in macrophages, participate in FcγR-mediated phagocytosis, integrin-dependent responses, and motility [73,74]. Full activation of the Arp2/3 complex requires nucleation-promoting factors from the WASP and WAVE protein families, which act as tumor suppressors or enhancers of malignant cells due to their multiple roles [75]. In sarcoma, we found upregulation of WASPs and FcγRs, suggesting anti-tumor functions. Within the circuit, the isoform PLCγ2, linked to the activation of WASPs, is positively correlated with good patient prognosis, and, in sarcoma, it promotes the infiltration of anti-tumor M1 macrophages, T cells, and monocytes into the TME [76]. 

A higher hazard ratio was found on a short circuit of the “Rap1 signaling pathway: calcium cation.” The circuit consists of TCR (T-cell receptor), the adaptor protein linker of activated T cells (LAT), and phospholipase Cγ1 (PLCG1). It is involved in the activation of T cells, resulting in the mobilization of calcium cations. In disease situations, overactive unstimulated TCRs induce changes in T cells to become unresponsive as a mechanism of immune evasion [77]. Furthermore, PLCγ1 is present in multiple signaling and disease pathways where it plays pro-tumorigenic roles, including a role in sarcomagenesis by a constitutively activated mutated PLCγ1 that promotes angiogenesis through activation of the VEGF pathway [78].

### 4.2. Roles in Metabolism

Sarcoma samples appear to have the energy metabolism wired towards gluconeogenesis, which, linked with the previous section on the TME, is relevant to maintaining anti-tumor immune conditions. Cellular energetics is also modulated by the NF-κΒ pathway, where the alternative NF-κΒ (RelB/p52 dimers) is activated by the IKK alpha dimer [79] to promote gluconeogenesis and oxidative metabolism instead of glycolysis, consistent with the downregulation of the IKK subtypes beta and gamma by some sarcoma subtypes. Increased activity of the gluconeogenic enzyme G6PC and decreased activity of the glycolysis-related glucose transporter GLUT1 mediated by the “adipocytokine signaling pathway” were observed. Adipocytokines affect the immune system, and it was found that DDLPS reduces its ability to produce adiponectin, which plays a role in suppressing macrophage function.

Protective roles are predicted for glucose metabolism mediators on the insulin and HIF-1 signaling pathways through the PI3K/Akt network. Activation of the “insulin signaling pathway” results in the inhibition of FOXO1 by phosphorylation. FOXO1 is a well-known TF that, in association with the coactivator PGC-1α (*PPARGC1A*), activates the gluconeogenic enzyme fructose 1,6-bisphosphatase (*FBP*), a putative inhibitor of sarcoma growth [80]. In contrast, HIF-1α regulates the transcription of genes encoding glycolytic enzymes, including the aldolase isozymes (*ALDOA*, *ALDOB*, and *ALDOC*). Previous research revealed a positive association of *ALDOB* expression with the survival of patients with gastric cancer [81] and *ALDOC* as a target for positive regulation by the retinoblastoma (RB) tumor suppressor protein [82]. Furthermore, *ALDOA* correlates positively with the immune infiltration of macrophages, neutrophils, and T cells [83].

### 4.3. Roles in Cell Motility

The preparation of tumor cells for migration includes cell-cell detachment accompanied by a change in cell morphology. In the first step, integrins and the focal adhesion kinase (*FAK* or protein tyrosine kinase 2, *PTK2*) receive signaling from cell attachment to neighbor cells. When detachment occurs, it triggers a specific type of apoptosis, called anoikis [84]. For pre-metastatic cells to overcome this barrier, multiple proteins have been associated with anoikis suppression, including the BCL-2 protein overactivated by the “focal adhesion” pathway. In the second step, a negative association with survival is obtained in the “axon guidance pathway: PTK2” (Table 1), a circuit related to ephrin-B reverse signaling where the activation of the ephrin-B ligands (*EFNB*) by the EPH receptors triggers a signaling cascade that increases FAK catalytic activity. This cascade causes the cells to modify their morphology into a round shape by remodeling the actin cytoskeleton and leading to cell repulsion [85]. In TCGA sarcoma data, higher expression of ephrin B1 and B2 ligands associated with enhancement of metastatic potential was found [86].

### 4.4. Roles in Cell Survival and Proliferation

Tumor cells usually acquire mechanisms to prevent cell death and enhance proliferation. Thus, it is no surprise that the “cell cycle pathway: RB1” (retinoblastoma protein as effector) signaling circuit (Table 1) was found to be associated with a low hazard ratio since the protein Rb, classified as a tumor suppressor, is activated by dephosphorylation to arrest the cell cycle [87]. Mutations or copy-number alterations on this pathway have been related to the development of sarcoma [37,87]. This is a short circuit consisting of three proteins. One of them, ARF, is a negative regulator of MDM2, which, in turn, negatively regulates Rb. The first gene is *CDKN2A* (a.k.a. *INK4A/ARF* locus), which encodes the proteins p16 and ARF that control the Rb and p53 pathways, respectively, with p16 being an inhibitor of the kinases that phosphorylate Rb and leading to growth arrest [88]. The second gene encodes murine double minute 2 (*MDM2*), a negative regulator of p53/Rb proteins inhibited by ARF [87,89]. *MDM2* is reported as an oncogene and driver alteration of liposarcoma [6], being coherently amplified in DDLPS expression data. 

On the other hand, the anti-apoptotic protein Bcl-2 appears to be overactivated by the “PI3K-Akt signaling pathway.” As seen before, this pathway is key in carcinogenesis, making *PI3K* a druggable target for anticancer therapy due to its role in the stimulation of cell growth and proliferation. Bcl-2 is a relevant driver of synovial sarcoma (SS) [90], being overexpressed in SS in TCGA data. The survival of tumor cells also depends on maintaining pro-proliferative ion homeostasis. A decreased angiotensin II signaling transmission leads to the under-activation of inositol 1,4,5-trisphosphate receptor calcium channels (IP3R by *IPTR* genes) and over-activation of TWIK-related acid-sensitive potassium channels (TASK by *KCNK* genes) mediated by the pathway “Aldosterone Synthesis and Secretion”. Excessive calcium release from IP3Rs disrupts mitochondrial membrane integrity and leads to apoptosis, a process suppressed by the anti-apoptotic Bcl-2 protein [91]. Meanwhile, TASK potassium channels are essential for cell survival with outward rectifying currents to maintain polarization of the cell [92,93]; overexpression of these channels has been associated with enhanced tumor cell proliferation and inhibition of apoptosis [93].

### 4.5. Specific Circuits in Sarcoma Subtypes

Besides common pathways that are generally dysregulated in sarcoma, there are also pathways specifically dysregulated in each sarcoma subtype (Table 5). Here, we explore the consequences of dysregulation of these circuits and the therapeutic actions that are taking place.

The specifically dysregulated circuits observed in DDLPS are closely related to metabolic alterations. Thus, the activation of the hedgehog signaling pathway, previously reported in sarcoma patients, would promote a dedifferentiated morphology of adipocytes, decreased lipid accumulation, and insulin resistance [94,95]. Increased insulin modulates the other two circuits seen in DDLPS, activating the “Regulation of lipolysis in adipocytes: PRKACA” circuit and suppressing the “Insulin signaling pathway: PCK1” circuit, respectively. Therapy targeting the hedgehog pathway has reduced efficacy, but co-targeting of interacting pathways, such as PI3K-AKT, may improve the therapeutic effect [94]. PI3K-AKT intervenes in signaling by insulin, protecting against glucose starvation and anoikis through the induction of the protein kinase A [96] and reduction in the triglyceride/fatty acid cycle fueling glyceroneogenesis in adipose tissue by reducing the activation of phosphoenolpyruvate carboxykinase 1 [97]. These changes suggest the metabolic rewiring of tumor cells towards survival in an undifferentiated and detached state while utilizing carbohydrate intermediates for biosynthesis instead of energy storage.

Specific responses observed in MFS are varied, including metabolic-, motility-, apoptosis-, and immune-related circuits. Increased uptake of glucose by tumor cells would raise the ATP/ADP ratio, leading to the inhibition of ATP-sensitive potassium channels and triggering a cascade to secrete insulin. In glioma cells, lower expression of *ABCC8* (encoding SUR1, a subunit of these channels) is associated with a poor prognosis [98]. Inhibition of these channels causes membrane depolarization, which is compensated by extracellular calcium intake. Among the multiple functions of calcium signaling is actin polymerization at the leading edge of migrating cells, further supported by the activation of the regulation of the actin cytoskeleton, which leads to actin branching on lamellipodia (Arp2/3) and inhibition of focal adhesion assembly (VCL, ACTN). Moreover, reducing tight junctions further supports tumor cell dissociation, high motility, and the invasive potential of MFS cells. On the other hand, we see the activation of the “natural killer cell-mediated cytotoxicity: KLRC4-KLRK1” circuit, involving the activating NK receptor NKG2D, which is a good prognostic marker on the TME and may be subject to cell immunotherapy [99]. 

In MPNST, significant differential activation in circuits related to the protective effect triggered by the TME on the FceRI signaling pathway, previously described, as well as a putative melanotic feature and neuron response to the hormones prolactin and oxytocin, was found. Different receptors activate Phospholipase C, triggering, in one case, melamin synthesis, previously observed in malignant melanotic nerve sheath tumors [100], and, in the oxytocin pathway, the release of calcium stored in intracellular pools into the cytoplasm via *RYR1*, leading to the secretion of oxytocin. In both cases, the prolactin and oxytocin pathways activate the MAPK signaling pathway to enhance cell proliferation through cyclin D1. These hormones induce tumor cell viability, and targeting their receptors may pose a treatment [101,102].

In SS, the down-activation of circuits with anticancer properties was mostly found. SS is related to MPNST, having a neural origin [103]. As such, dysregulation of the endocannabinoid system is observed, which impacts multiple cancer-related pathways with anti-tumor effects concurrently with the down-activation of SS [104]. In terms of metabolism, the glucagon signaling pathway down-activates glycogenolysis and glycolysis by GYS and PKM, respectively, which are circuits that exhibit differential behaviors depending on the cancer type. At the same time, it activates fatty acid oxidation by CPT1, which is involved in the metabolic adaptation of tumor cells to produce energy and metabolic intermediates to promote cancer progression [105]. In concordance, the Jak-STAT signaling pathway inactivates p21, which is involved with cell-cycle inhibition. 

In STLMS, reduced signaling from tumor necrosis factor (TNF) in the adipocytokine signaling pathway leads to elevated activation of IRS, involved in insulin sensitivity, whose expression has been related to malignant sarcoma progression. This protein was also found to be constitutively activated in leiomyosarcoma [106]. Unexpectedly, a down-activation of the oncoprotein MYC mediated by the PI3K-Akt signaling pathway was observed. Given the limited number of cases available in the dataset, this circuit may be altered by current patient treatments.

Dysregulated signaling in the VEGF signaling pathway in ULMS, putatively produced from the reduced expression of the VEGFA ligand, leads to increased anti-apoptotic *BAD* and *CASP9* and decreased *COX-2* activation. The production of VEGF promotes a mechanism to evade the immune system by inhibiting the migration of T cells after modification of gene expression with a reduction in adherent factors in endothelial cells [77].

UPS overactivates circuits involved in cell proliferation, such as the JAK/STAT signaling pathway in response to chemokine, with the production of complement anaphylatoxins. These anaphylatoxins play a role in regulating inflammation, enhancing cell stemness, and activating the CRAC (Ca^2+^ release-activated Ca^2+^) channel ORAI1, a potential contributor to cancer progression. On the other hand, under-activation of the antiproliferative GnRH signaling pathway via JNK/c-jun signaling was observed. Therapeutic anticancer treatments over these pathways include complement-dependent cytotoxicity with antibodies and the utilization of GnRH agonists to counteract the signaling of growth-factor receptors or antagonists to induce apoptosis [107].

### 4.6. Alternative Validations

Since the analysis presented here comprises the transmission of signaling activity throughout proteins involved in the pathways, those proteins that may have a relevant impact on patient survival were further analyzed with other alternative tools (KMplot and UALCAN) for additional support. Although the results provided by KMplot and UALCAN are obtained from different cancers, Appendix A shows a high level of concordance between both results, providing extra validation of the results produced by the model. 

### 4.7. Clinical Strategies

Chemotherapy and targeted therapy are both potent methods to treat cancer. However, due to their different forms of action, the side effects vary. While chemotherapeutics affect cells in the body with rapid proliferation, they affect not only tumor cells but also cells in the bone marrow and hair. On the other hand, targeted therapy agents, usually small molecules or antibodies directed towards specific proteins, have reduced side effects that commonly include diarrhea and skin reactions. The study of signaling pathways allows for the detection of putative targets on the pathways, either on specific nodes or in adjacent nodes, that may contribute to the programmed therapeutic perturbation. Many circuits share alterations in the PI3K/Akt/mTOR and Ras/MAPK networks. Both signaling networks may be activated by multiple RTKs, for which numerous tyrosine kinase inhibitors (TKIs) are sarcoma-approved drugs like pazopanib, sunitinib, sorafenib, or erlotinib [108]. 

Besides signaling pathway modulation, immunotherapy is a promising field for the treatment of soft tissue sarcoma patients. As previously discussed, our results indicate that an enhanced TME is usually a sign of a good patient prognosis. A challenge is to promote an anti-tumor TME and reduce resistance to therapy. The adoptive cell therapy of T-cells consists of manipulating the TCR to recognize tumor-specific antigens and promote an anti-tumor response in the patient [109]. Another strategy is to use molecules to prevent polarization from the anti-tumor M1 phenotype to the immunosuppressive M2 phenotype of tumor-associated macrophages. In this work, various circuits within Fc gamma R-mediated phagocytosis, the adipocytokine signaling pathway, and the HIF-1 signaling pathway (see Table 5) related to macrophage polarization that may be considered for therapy were discussed. Furthermore, monoclonal antibodies could be designed to block checkpoint receptors like CTLA-4 or PD-1, which are more abundant in non-translocation-associated sarcomas (such as six of the TCGA subtypes analyzed here, excluding SS).

## 5. Conclusions

Sarcoma tumors are difficult to treat, leading to poor overall survival for patients. In the present work, the activation of pathways that have a significant association with patient survival and those that are dysregulated in tumor samples compared to normal tissues was assessed. Traditional neoadjuvant therapies usually target known oncogenic circuits, while newer therapies are focused on improving the immune anticancer response. In that respect, the protective role of signaling and metabolic pathways that promote the infiltration of anti-tumorigenic immune cells into the sarcoma TME has been discussed. On the other hand, different tumorigenic functions, such as cytoskeleton remodeling and the promotion of tumor and pre-metastatic cell survival, were found to negatively affect patient outcomes. Interestingly, the same elements may be indicative of a good or bad prognosis depending on the specific pathway and cell type they participate in. A systematic review has been conducted, taking into consideration the high variability of cellular processes existent in different types of sarcoma. A relevant outcome was the identification of common circuits that potentially influence patient survival and tumor progression in soft tissue sarcoma. The findings presented here provide hints for new therapeutic interventions in sarcoma.

## Figures and Tables

**Figure 1 ijms-24-14732-f001:**
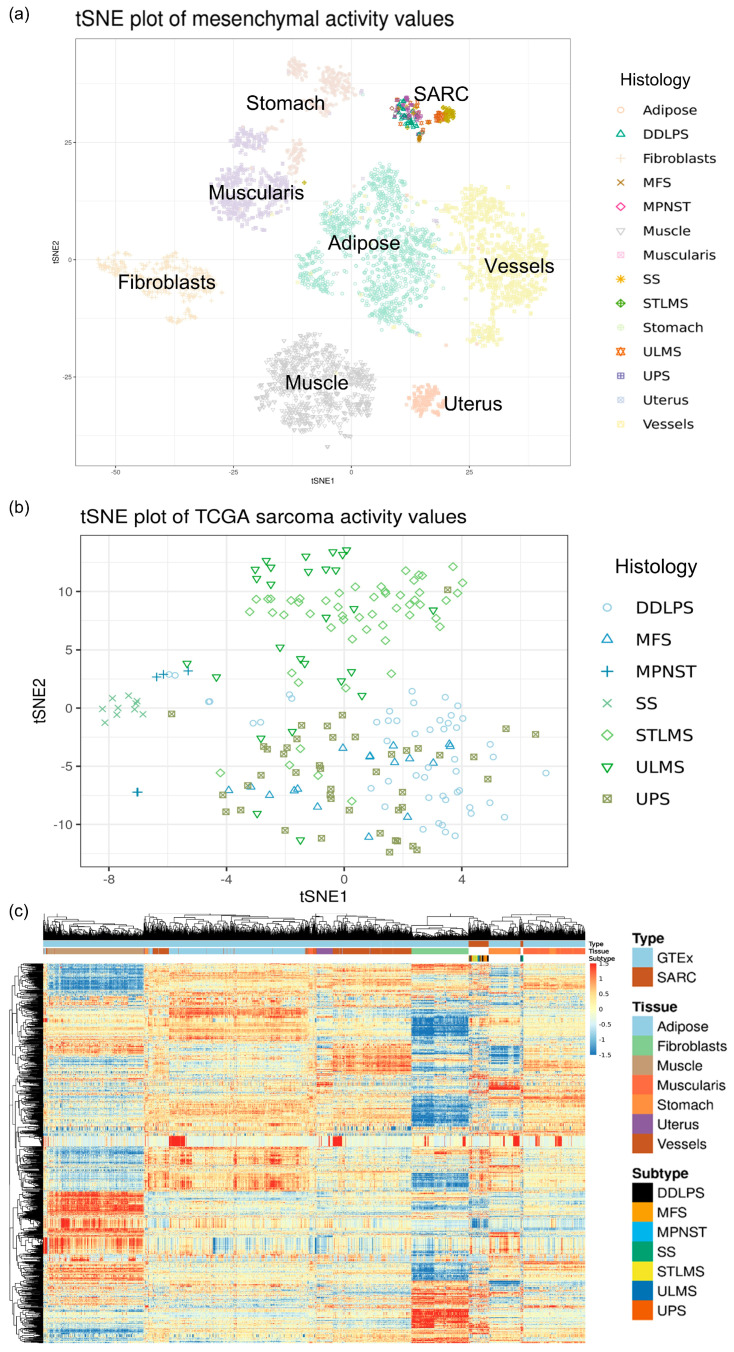
Clustering of sarcoma and normal samples from normalized circuit activity values. tSNE plots with samples from all (**a**) or only sarcoma (**b**) projects and a heatmap with sample clustering using correlation distance (**c**) for all sarcoma and normal samples analyzed.

**Figure 2 ijms-24-14732-f002:**
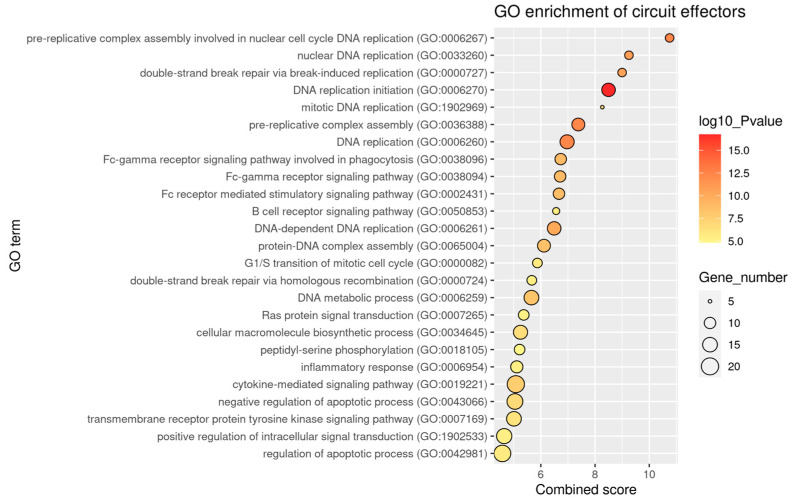
Enriched Gene Ontology terms on the effector genes of the 108 circuits that are differentially activated in 5 or more sarcoma subtypes. Biological process ontology is shown.

**Figure 3 ijms-24-14732-f003:**
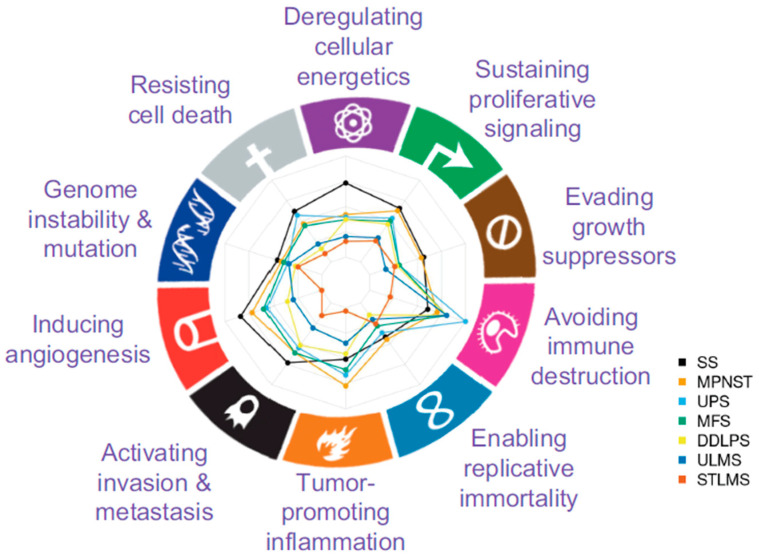
Distribution of Hallmarks of Cancer on the sets of differentially activated circuits for each sarcoma subtype. Along the axis of the radar plot, icons representing the cancer hallmarks, as depicted in Hanahan and Weinberg [32], were included.

**Table 1 ijms-24-14732-t001:** Circuits in soft tissue sarcoma have a significant association with the overall survival of patients. Hazard ratios (HR) were obtained by analyzing 206 sarcoma patients with Cox proportional hazards models.

Pathway Name (KEGG ID)	Circuit Effector Node	HR (95% CI for HR)	Concordance	FDR ^1^	p.zph ^2^
Rap1 signaling pathway (hsa04015)	Calcium cation	2.7 × 10^29^ (4.1 × 10^17^–1.77 × 10^41^)	0.663	0.00116	0.426
Fc epsilon RI signaling pathway (hsa04664)	FCER1G MS4A2 FCER1A	2.06 × 10^−20^ (2.74 × 10^−29^–1.54 × 10^−11^)	0.645	0.00474	0.603
Fc epsilon RI signaling pathway (hsa04664)	AKT3	1.60 × 10^−3^ (8.49 × 10^−5^–3.02 × 10^−2^)	0.667	0.00474	0.343
Fc epsilon RI signaling pathway (hsa04664)	MAPK8	5.02 × 10^−3^ (4.83 × 10^−4^–5.22 × 10^−2^)	0.664	0.00474	0.412
Fc epsilon RI signaling pathway (hsa04664)	MAPK14	5.21 × 10^−3^ (4.49 × 10^−4^–6.04 × 10^−2^)	0.649	0.00573	0.526
Insulin signaling pathway (hsa04910)	FBP1	6.21 × 10^−7^ (5.68 × 10^−10^–6.79 × 10^−4^)	0.644	0.0106	0.323
Fc epsilon RI signaling pathway (hsa04664)	PLA2G4B	5.83 × 10^−225^ (0.0–1.21 × 10^−114^)	0.627	0.0106	0.616
Platelet activation (hsa04611)	Thromboxane A2	2.00 × 10^−3^ (9.04 × 10^−5^–4.43 × 10^−2^)	0.655	0.0115	0.045
Cell cycle (hsa04110)	RB1	2.26 × 10^−4^ (3.13 × 10^−6^–1.63 × 10^−2^)	0.635	0.0146	0.198
Axon guidance (hsa04360)	PTK2	1.63 × 10^7^ (2.91 × 10^3^–9.11 × 10^10^)	0.611	0.0178	0.591
MAPK signaling pathway (hsa04010)	NFKB1	2.94 × 10^−14^ (6.33 × 10^−22^–1.36 × 10^−6^)	0.62	0.0481	0.375
HIF-1 signaling pathway (hsa04066)	ALDOA	3.66 × 10^−8^ (2.15 × 10^−12^–0.000622)	0.62	0.0481	0.440
Insulin signaling pathway (hsa04910)	PPARGC1A	6.98 × 10^−155^ (2.62 × 10^−242^–1.86 × 10^−67^)	0.632	0.0481	0.220

^1^ False discovery rate (FDR) was calculated with the method of Benjamini and Hochberg on the *p*-values of the Cox function for all circuits. ^2^ Adjusted *p*-values of the proportional hazards assumption calculated with the *cox.zph* function.

**Table 2 ijms-24-14732-t002:** Set of transcription factors is generally activated in soft tissue sarcoma. Regulatory roles in cancer from the TSGene, ONGene, and COSMIC databases are added.

Symbol	Name	TSG/ONG	COSMIC	n	LOR ^1^	FDR ^2^
*FOXD1*	Forkhead Box D1			7	0.43	1.05 × 10^−2^
*SPI1*	Spi-1 Proto-Oncogene	both		6	0.09	1.35 × 10^−5^
*GATA3*	GATA Binding Protein 3		both	6	0.09	6.23 × 10^−4^
*IKZF1*	IKAROS Family Zinc Finger 1	both	TSG	6	0.27	3.01 × 10^−3^
*MAF*	MAF BZIP Transcription Factor	ONG	ONG	6	0.26	9.55 × 10^−3^
*RFX5*	Regulatory Factor X5			6	0.55	7.21 × 10^−3^
*TCF4*	Transcription Factor 4	TSG		6	0.11	4.53 × 10^−3^
*ZEB2*	Zinc Finger E-Box Binding Homeobox 2			6	0.24	4.46 × 10^−4^

^1^ Average log odds ratio (LOR) indicates the average enrichment of transcription factor targets across the indicated number of subtypes with a significant result. ^2^ Average false discovery rate (FDR) was calculated with the method of Benjamini and Hochberg and averaged across the significant subtypes.

**Table 3 ijms-24-14732-t003:** Differentially activated circuits in the seven sarcoma subtypes are subject to analysis with respect to non-diseased sarcomagenic GTEx tissues.

Pathway Name (KEGG ID)	Circuit Effector	log2FC ^1^	FDR ^2^
Aldosterone synthesis and secretion (hsa04925)	ITPR1	−0.101	1.87 × 10^−3^
Fc gamma R-mediated phagocytosis (hsa04666)	WAS	0.081	2.86 × 10^−3^
Fc gamma R-mediated phagocytosis (hsa04666)	ARPC5	0.089	4.24 × 10^−3^
Adipocytokine signaling pathway (hsa04920)	SLC2A1	−0.245	4.45 × 10^−3^
Adipocytokine signaling pathway (hsa04920)	Long-chain fatty acid	−0.184	4.95 × 10^−3^
Aldosterone synthesis and secretion (hsa04925)	KCNK3	0.089	8.12 × 10^−3^
Focal adhesion (hsa04510)	BCL2	0.072	9.94 × 10^−3^
Adipocytokine signaling pathway (hsa04920)	G6PC	0.174	1.22 × 10^−2^

^1^ Average log2 of the fold change obtained with the limma pipeline contrasting tumor and normal samples and averaging across the seven sarcoma subtypes with significant results in the same regulatory direction. ^2^ Average false discovery rate (FDR) was calculated with the method of Benjamini and Hochberg and averaging for the seven subtypes.

**Table 4 ijms-24-14732-t004:** Description of the contrasted sarcoma and normal tissues from the TCGA SARC and GTEx projects.

Sarcoma	Sarcoma Descriptions	GTEx	GTEx Project	Tissue Descriptions
DDLPS	Dedifferentiated Liposarcoma	Adipose	Adipose Tissue	Adipose—Subcutaneous
ULMS	Uterine Leiomyosarcoma	Uterus	Uterus	Uterus
STLMS	Soft Tissue Leiomyosarcoma	Muscularis	Esophagus	Esophagus—Muscularis
STLMS	Soft Tissue Leiomyosarcoma	Stomach	Stomach	Stomach
STLMS	Soft Tissue Leiomyosarcoma	Vessels	Blood Vessel	Artery—Coronary; Artery—Tibial
MPNST	Malignant Peripheral Nerve Sheath Tumors (MPNST)	Fibroblasts	Skin	Cells—Cultured fibroblasts
MFS	Myxofibrosarcoma	Fibroblasts	Skin	Cells—Cultured fibroblasts
MFS	Myxofibrosarcoma	Adipose	Adipose Tissue	Adipose—Subcutaneous
MFS	Myxofibrosarcoma	Vessels	Blood Vessel	Artery—Coronary; Artery—Tibial
MFS	Myxofibrosarcoma	Muscle	Muscle	Muscle—Skeletal
MFS	Myxofibrosarcoma	Muscularis	Esophagus	Esophagus—Muscularis
UPS	Undifferentiated Pleomorphic Sarcoma	Fibroblasts	Skin	Cells—Cultured fibroblasts
UPS	Undifferentiated Pleomorphic Sarcoma	Muscularis	Esophagus	Esophagus—Muscularis
SS	Synovial Sarcoma	Fibroblasts	Skin	Cells—Cultured fibroblasts

**Table 5 ijms-24-14732-t005:** Uniquely dysregulated pathways by each specific sarcoma subtype.

Subtype	Pathway Name	Circuit Effector	Dysregulation
DDLPS	Hedgehog signaling pathway	SMO	up
DDLPS	Insulin signaling pathway	PCK1	down
DDLPS	Regulation of lipolysis in adipocytes	PRKACA	up
MFS	Gap junction	GJA1 TJP1	down
MFS	Insulin secretion	ABCC8	down
MFS	Natural killer cell-mediated cytotoxicity	KLRC4-KLRK1	up
MFS	NOD-like receptor signaling pathway	CASP8	up
MFS	Phospholipase D signaling pathway	D-myo-Inositol 1,4,5-trisphosphate	up
MFS	Regulation of actin cytoskeleton	ACTB ARPC5	up
MFS	Regulation of actin cytoskeleton	ACTN4	down
MFS	Regulation of actin cytoskeleton	VCL	down
MPNST	Fc epsilon RI signaling pathway	AKT3	up
MPNST	Fc epsilon RI signaling pathway	PLA2G4B	up
MPNST	Melanogenesis	CAMK2A	up
MPNST	Melanogenesis	PRKACA	up
MPNST	Oxytocin signaling pathway	RYR1	up
MPNST	Prolactin signaling pathway	CCND1	up
SS	ErbB signaling pathway	CBLC	down
SS	ErbB signaling pathway	MYC	down
SS	Glucagon signaling pathway	CPT1C	up
SS	Glucagon signaling pathway	GYS1	down
SS	Glucagon signaling pathway	PKM	down
SS	Jak-STAT signaling pathway	CDKN1A	down
SS	Retrograde endocannabinoid signaling	GRIA1	down
SS	Retrograde endocannabinoid signaling	PRKCA	down
STLMS	Adipocytokine signaling pathway	IRS1	up
STLMS	PI3K-Akt signaling pathway	MYC	down
ULMS	VEGF signaling pathway	BAD	up
ULMS	VEGF signaling pathway	CASP9	up
ULMS	VEGF signaling pathway	PTGS2	down
UPS	Chemokine signaling pathway	STAT1	up
UPS	Complement and coagulation cascades	C3	up
UPS	GnRH signaling pathway	JUN	down
UPS	GnRH signaling pathway	PLD1	down
UPS	Platelet activation	ORAI1	up

## Data Availability

Transcriptomics data belong to the public repository of Recount3 (https://doi.org/10.1186/s13059-021-02533-6, accessed on 1 September 2023). All the code and additional tables are available in the github repository: https://github.com/babelomics/TCGA_SARC, accessed on 1 September 2023. Updated sarcoma metadata performed for the TCGA SARC project by the TCGA Research Network was obtained from Appendix A at https://doi.org/10.1016/j.cell.2017.10.014, accessed on 1 September 2023. Publicly available datasets were used for annotation in this study: COSMIC (Catalogue Of Somatic Mutations In Cancer) v96, Cancer Gene Census: https://cancer.sanger.ac.uk/cosmic/file_download_info?data=GRCh38%2Fcosmic%2Fv96%2Fcancer_gene_census.csv, accessed on 1 September 2023. Oncogene database: https://ongene.bioinfo-minzhao.org/ongene_human.txt, accessed on 1 September 2023. Tumor Suppressor Gene database: https://bioinfo.uth.edu/TSGene/Human_TSGs.txt, accessed on 1 September 2023.

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
