# Peer review of "Functional Profiling of Soft Tissue Sarcoma Using Mechanistic Models"

_ijms, 2023, doi:10.3390/ijms241914732_

Round 1

Reviewer 1 Report

In this manuscript, Paya-Milans et al suggested a specific signaling-pathway signature associated with sarcoma patients’ survival. The authors followed a purely bioinformatics approach, using a variety of tools and techniques that adequately support their results. Despite the interest of the study and its outcomes, there are some points that need to be addressed:

1.    Authors very successfully detected and suggested specific nodes within the differentially regulated pathways and circuits as predictors of patients’ survival. Did the authors consider validating the potential of these molecules using other bioinformatics tools such as the UALCAN and/or KMplotter tools? It would be good to provide a table and figure with these data that would make the results more sound.

2.    Since Figure 3 seems to be an amended version of the well-known schematic figure by Hanahan and Weinberg, authors should cite their publication and mention in the figure legend that this figure is herein reproduced and edited. 、

3.    Names of genes should be typed in italics.  

Author Response

COMMENT

=========

1.    Authors very successfully detected and suggested specific nodes within the differentially regulated pathways and circuits as predictors of patients’ survival. Did the authors consider validating the potential of these molecules using other bioinformatics tools such as the UALCAN and/or KMplotter tools? It would be good to provide a table and figure with these data that would make the results more sound.

RESPONSE

=========

We appreciate very much the comment. We have followed the suggestion and generate a new table. In general, there is a good agreement between the predictions of the model and the values for the molecules present in the predicted circuits provided by the tools (even thought that there were not the same cancers). We thank the referee for this suggestion that has improved the quality of the manuscript.

COMMENT

=========

2.    Since Figure 3 seems to be an amended version of the well-known schematic figure by Hanahan and Weinberg, authors should cite their publication and mention in the figure legend that this figure is herein reproduced and edited. 、

RESPONSE

=========

This is a good suggestion, thanks. Done.

COMMENT

=========

3.    Names of genes should be typed in italics. 

RESPONSE

=========

Done

Reviewer 2 Report

The authors presented transcriptomics profiling of sarcoma using developed earlier mechanistic models. The manuscript fits to the journal topic on bioinformatics. It is well balanced computational study using RNA-seq data.

However, some updates are necessary. First, the term ‘mechanistic model’ is not widely used. It is worthy add some description of such model as mathematical model, or computational model on signal transduction based on correlations.

Please add it to the Abstract, rephrase like “We used previously developed computational approach called ‘mechanistic model’ to…”

Not just ‘Here, mechanistic models … have been employed…’ (line 17)

Then, the work is focused on sarcoma, not other types of cancers. Change or remove word ‘specifically’ (line 20)

The abbreviations STS and NCT are used only once in the Abstract. Remove abbreviations, then give it in the main text.

Concluding phrase in the Abstract is not concrete, too common – “deeper understanding …can be obtained … paving the way for potential…”. Write what is shown, how it will be used for sarcoma diagnosis, possible therapy, not just ‘understanding’

Add keywords ‘RNA-seq’, ‘transcriptome’, ‘profiling’ to the keywords.

Line 60: ‘different diseases [9-11]’ – name these diseases, and try not cite together 3 or more references.

Line 70: Need describe what ‘mechanistic model’ means here.

Rephrase ‘since mechanistic models convey causality…” – need prove it by a reference. In this paragraph (after line 62), the application of the mechanistic models are given, but not the idea of computations.

Line 80: ‘HiPathia’ – later in the text it is Hipathia. It could be written as HIPATHIA too. Please use same writing upper/lowercase letters through all the text. Show the abbreviation in full at first mention in the text (it is in line 110 now)

Line 90: ‘STAR and Megadepth’ – need references for these tools.

Line 96: ‘org.Hs.eg.db package’ – add URL or reference here.

Line 102: ‘M-values (TMM)’ – comment what is ‘M-value’

Line 117: ‘R package  Rtsne version 0.15’ – please rephrase – is it version 0.15 or the package, or R?

Line 123: ‘’ version 3.2-10’ – is it proper number? Or ‘version higher than…’ ?

Line 130: ‘cox.zph function’ – add a reference.

Line 133: ‘cutoff of +/- 0.5.’ – comment on the scale. Why this cut-off?

Line 144: name  TFTEA as a tool, or method. Add word.

Line 144: ‘dorothea regulons’ – comment what is Dorothea.

Then ‘From dorothea’ – add word database, or the link to the data, or reference.

‘(confidence levels A, B and C)’ – what mean these levels – A,B,C? please comment. IT is some internal parameter not clear to the reader.

Line 150: ‘ONGENE and TSGENE.’ – give references to these databases, abbreviations in full.

Line 159: ‘0.15201’ – why this value, with such precision, what is the scale? Change the phrase, comment how the cut-off was selected.

Figure 1 has low resolution. It is hard to see panels (b) and (c)

Hard to distinguish colors marking tissues. Fong size is too small to read.

Is it possible to make panel (c) in page width?

Line 186: ‘as defined in the  KEGG  repository’ – rephrase, remove ‘as’, give KEGG abbreviation in full. It is about specific repository in KEGG, or the database as whole?

t

Table 2. Add word (pathway ID) to the title of first column ‘Pathway name’. Or add comment to the Table Note – where such IDs like (hsa04015) from?

Column HR in the Table – use same scale (exponent and linear) for the numbers, avoid breaking lines in exponent.

Line 264: ‘Terms on the Biological Process ontology are shown.’ – remove it from the figure legend, put word ‘Biological Process’ to first sentence in the Figure legend

Line 267 and below – add wording ‘Table Note’, and make smaller font for the table note (comments) 1 and 2, or mark it by font, not add empty lines.

Same comment is for other Tables. Mark ‘Table Note’ comments by font, show it explicitly.

Figure 3 ‘’ Hallmarks of Cancer’ is interesting. But the central diagram (percent) is too small, the lines are thin. It is hard to distinguish colors. Please update, make it larger, make lines more visible (thick)

Line 310: ‘in cancer, including sarcoma  [45-47].’ – mention cancer types explicitly.

Line 518: ‘In this work, various circuits’ – change word ‘various’, name these circuits.

Line 520: ‘monoclonal antibodies are designed’ – designed where? Add a reference here.

In conclusion – avoid common phrases ‘A comprehensive review has been conducted’ (line 535)

Rephrase.

See also “The findings presented here paved the way for further research, offering promising…”

Please write the sentence in more precise and less ambitious manner.

In the Acknowledgements some phrases are redundant (Institutional Review Board Statement) – may just write ‘N/A’ (on the authors’ discretion)

Author Response

Comments and Suggestions for Authors

The authors presented transcriptomics profiling of sarcoma using developed earlier mechanistic models. The manuscript fits to the journal topic on bioinformatics. It is well balanced computational study using RNA-seq data.

COMMENT

=========

However, some updates are necessary. First, the term ‘mechanistic model’ is not widely used. It is worthy add some description of such model as mathematical model, or computational model on signal transduction based on correlations.

RESPONSE

=========

We agree with the referee and we have included a description where requested, eight comments below.

COMMENT

=========

Please add it to the Abstract, rephrase like “We used previously developed computational approach called ‘mechanistic model’ to…”

Not just ‘Here, mechanistic models … have been employed…’ (line 17)

RESPONSE

=========

Done

COMMENT

=========

Then, the work is focused on sarcoma, not other types of cancers. Change or remove word ‘specifically’ (line 20)

RESPONSE

=========

Removed

COMMENT

=========

The abbreviations STS and NCT are used only once in the Abstract. Remove abbreviations, then give it in the main text.

RESPONSE

=========

Done

COMMENT

=========

Concluding phrase in the Abstract is not concrete, too common – “deeper understanding …can be obtained … paving the way for potential…”. Write what is shown, how it will be used for sarcoma diagnosis, possible therapy, not just ‘understanding’

RESPONSE

=========

Since what we have found and what is useful for was described above in the abstract, we have just removed the sentence, that simply aimed to be a general corollary but, as the referee noted, not concrete.

COMMENT

=========

Add keywords ‘RNA-seq’, ‘transcriptome’, ‘profiling’ to the keywords.

RESPONSE

=========

Done

COMMENT

=========

Line 60: ‘different diseases [9-11]’ – name these diseases, and try not cite together 3 or more references.

RESPONSE

=========

We have listed the diseases corresponding to the publications (cancer, diabetes and Fanconi Anemia)

COMMENT

=========

Line 70: Need describe what ‘mechanistic model’ means here.

RESPONSE

=========

We have described the concept of mechanistic model and added in methods a more extended section in which we explain how the mechanistic model used here works.

COMMENT

=========

Rephrase ‘since mechanistic models convey causality…” – need prove it by a reference. In this paragraph (after line 62), the application of the mechanistic models are given, but not the idea of computations.

RESPONSE

=========

We have rewritten the sentence. We think that now is more self-explanatory and increases the clarity of the text.

COMMENT

=========

Line 80: ‘HiPathia’ – later in the text it is Hipathia. It could be written as HIPATHIA too. Please use same writing upper/lowercase letters through all the text. Show the abbreviation in full at first mention in the text (it is in line 110 now)

RESPONSE

=========

Apologies. We have unified the acronym to HiPathia, as in the Bioconductor/R package

COMMENT

=========

Line 90: ‘STAR and Megadepth’ – need references for these tools.

RESPONSE

=========

Added

COMMENT

=========

Line 96: ‘org.Hs.eg.db package’ – add URL or reference here.

RESPONSE

=========

Added

COMMENT

=========

Line 102: ‘M-values (TMM)’ – comment what is ‘M-value’

RESPONSE

=========

The sentence was a bit confusing and out of context. We have rewritten the sentence to “The downloaded raw counts for all samples were normalized using the Trimmed Mean of M-values (TMM) method from edgeR package version 3.32.1 [31], as implemented in the edgeR package [32] version 3.32.1.” The M-values are the log-expression ratios extensively used in gene expression normalization

COMMENT

=========

Line 117: ‘R package  Rtsne version 0.15’ – please rephrase – is it version 0.15 or the package, or R?

RESPONSE

=========

It is the version of the package. We have removed the R to avoid confusion. We have also added a reference to the package.

COMMENT

=========

Line 123: ‘’ version 3.2-10’ – is it proper number? Or ‘version higher than…’ ?

RESPONSE

=========

Although it may look weird it is the way in which the author quoted the version.

COMMENT

=========

Line 130: ‘cox.zph function’ – add a reference.

RESPONSE

=========

It is a function of the survival package (mentioned at the beginning of the paragraph with the corresponding reference). We have clarified this point in the text. 

COMMENT

=========

Line 133: ‘cutoff of +/- 0.5.’ – comment on the scale. Why this cut-off?

RESPONSE

=========

We have reformulated the sentence because it was a bit confusing. We meant that high and low activity was defined and the upper and lower 25% percentiles of the observed values.

COMMENT

=========

Line 144: name  TFTEA as a tool, or method. Add word.

RESPONSE

=========

We realized that it is a bit confusing. Actually, the method and the tool have the same name, so we have rewritten the sentence to: “Results of differential expression analysis were used for indirect estimation of transcription factor activity by enrichment analysis of their corresponding target genes using the Transcription Factor Target Enrichment Analysis (TFTEA) [27] tool.” for the sake of clarity.

COMMENT

=========

Line 144: ‘dorothea regulons’ – comment what is Dorothea.

RESPONSE

=========

We have rewritten the sentence indicating what Dorotea resource is. Now it reads: “The set of transcription factor-target interactions used in this analysis was obtained from the papers describing TFTEA [27] and Dorothea, a database that collects transcription factor’s targets (regulons) [38]. We think that now the meaning is clearer.

COMMENT

=========

Then ‘From dorothea’ – add word database, or the link to the data, or reference.

RESPONSE

=========

Done

COMMENT

=========

 ‘(confidence levels A, B and C)’ – what mean these levels – A,B,C? please comment. IT is some internal parameter not clear to the reader.

RESPONSE

=========

These confidence levels are defined in the Dorothea publication. We have explicitly mentioned this in the text.

COMMENT

=========

Line 150: ‘ONGENE and TSGENE.’ – give references to these databases, abbreviations in full.

RESPONSE

=========

Added

COMMENT

=========

Line 159: ‘0.15201’ – why this value, with such precision, what is the scale? Change the phrase, comment how the cut-off was selected.

RESPONSE

=========

We have reformulated the phrase explaining the origin of the cut-off. And also, we removed some non-significant decimals from the cut-off.

COMMENT

=========

Figure 1 has low resolution. It is hard to see panels (b) and (c)

Hard to distinguish colors marking tissues. Fong size is too small to read.

Is it possible to make panel (c) in page width?

RESPONSE

=========

 We have prepared a new version of the figure solving all the problems mentioned by the referee.

COMMENT

=========

Line 186: ‘as defined in the  KEGG  repository’ – rephrase, remove ‘as’, give KEGG abbreviation in full. It is about specific repository in KEGG, or the database as whole?

RESPONSE

=========

Done. It is about the repository of pathways.

COMMENT

=========

Table 2. Add word (pathway ID) to the title of first column ‘Pathway name’. Or add comment to the Table Note – where such IDs like (hsa04015) from?

RESPONSE

=========

They are KEGG IDs. We have included this information in the column header, as requested by the referee.

COMMENT

=========

Column HR in the Table – use same scale (exponent and linear) for the numbers, avoid breaking lines in exponent.

RESPONSE

=========

Done

COMMENT

=========

Line 264: ‘Terms on the Biological Process ontology are shown.’ – remove it from the figure legend, put word ‘Biological Process’ to first sentence in the Figure legend

RESPONSE

=========

Done

COMMENT

=========

Line 267 and below – add wording ‘Table Note’, and make smaller font for the table note (comments) 1 and 2, or mark it by font, not add empty lines.

RESPONSE

=========

Done

COMMENT

=========

Same comment is for other Tables. Mark ‘Table Note’ comments by font, show it explicitly.

RESPONSE

=========

Done

COMMENT

=========

Figure 3 ‘’ Hallmarks of Cancer’ is interesting. But the central diagram (percent) is too small, the lines are thin. It is hard to distinguish colors. Please update, make it larger, make lines more visible (thick)

RESPONSE

=========

 We have prepared a new larger version of the figure with the lines more visible. In any case, please, see the high-quality version of the figure uploaded with the manuscript, because the figure embedded in the text may lose quality in the word document.

COMMENT

=========

Line 310: ‘in cancer, including sarcoma  [45-47].’ – mention cancer types explicitly.

RESPONSE

=========

We have listed the cancers as requested.

COMMENT

=========

Line 518: ‘In this work, various circuits’ – change word ‘various’, name these circuits.

RESPONSE

=========

We have mentioned the circuits (Fc gamma R-mediated phagocytosis, adipocytokine signaling pathway and HIF-1 signaling pathway) as requested.

COMMENT

=========

Line 520: ‘monoclonal antibodies are designed’ – designed where? Add a reference here.

RESPONSE

=========

This is a typo. In the paragraph we were suggesting possible strategies and we meant that antibodies “could be” designed… We have fixed the typo. Now it reads “…monoclonal antibodies could be designed…”

COMMENT

=========

In conclusion – avoid common phrases ‘A comprehensive review has been conducted’ (line 535)

Rephrase.

See also “The findings presented here paved the way for further research, offering promising…”

Please write the sentence in more precise and less ambitious manner.

RESPONSE

=========

We have rewritten the sentences as requested by the referee. We hope that the text now look less ambitious.

COMMENT

=========

In the Acknowledgements some phrases are redundant (Institutional Review Board Statement) – may just write ‘N/A’ (on the authors’ discretion)

RESPONSE

=========

For the acknowledgement we textually copied the sentences from the sources, written the way they require to be cited. We have written N/A in the institutional board review as requested.